# Genetic and Genomic Tools in Sunflower Breeding for Broomrape Resistance

**DOI:** 10.3390/genes11020152

**Published:** 2020-01-30

**Authors:** Sandra Cvejić, Aleksandra Radanović, Boško Dedić, Milan Jocković, Siniša Jocić, Dragana Miladinović

**Affiliations:** Institute of Field and Vegetable Crops, 21000 Novi Sad, Serbia; aleksandra.dimitrijevic@ifvcns.ns.ac.rs (A.R.); bosko.dedic@ifvcns.ns.ac.rs (B.D.); milan.jockovic@ifvcns.ns.ac.rs (M.J.); sinisa.jocic@ifvcns.ns.ac.rs (S.J.); dragana.miladinovic@ifvcns.ns.ac.rs (D.M.)

**Keywords:** sunflower, broomrape, resistance, genes, genome

## Abstract

Broomrape is a root parasitic plant causing yield losses in sunflower production. Since sunflower is an important oil crop, the development of broomrape-resistant hybrids is the prime breeding objective. Using conventional plant breeding methods, breeders have identified resistant genes and developed a number of hybrids resistant to broomrape, adapted to different growing regions worldwide. However, the spread of broomrape into new countries and the development of new and more virulent races have been noted intensively. Recent advances in sunflower genomics provide additional tools for plant breeders to improve resistance and find durable solutions for broomrape spread and virulence. This review describes the structure and distribution of new, virulent physiological broomrape races, sources of resistance for introduction into susceptible cultivated sunflower, qualitative and quantitative resistance genes along with gene pyramiding and marker assisted selection (MAS) strategies applied in the process of increasing sunflower resistance. In addition, it presents an overview of underutilized biotechnological tools, such as phenotyping, -omics, and genome editing techniques, which need to be introduced in the study of sunflower resistance to broomrape in order to achieve durable resistance.

## 1. Introduction

Sunflower (*Helianthus annuus* L.) is the fourth most important oil crop in the world, mainly grown in temperate, semi-dry regions. In addition to its primary intention for human consumption, sunflower oil has a wide range of applications and can be used as a supplement in chemical as well as pharmaceutical industries [1]. The worldwide production of sunflower is harmed by many biotic constraints, among which the parasitic weed broomrape (*Orobanche cumana* Wallr.) is one of the most devastating. 

Broomrape is present in many countries of Europe and Asia, especially in Central and Eastern Europe, Spain, Turkey, Israel, Iran, Kazakhstan, and China [2]. Over the past few years the progression of broomrape, its introduction into new countries, and the development of new and more virulent races have been under intensive observation [3]. Sunflower broomrape has a great capacity for dispersion and mutation [4]. Individual broomrape plants produce thousands of minute seeds that are easily dispersed by wind and other agents, including sunflower seeds, to which broomrape seed can be attached. Broomrape seed may remain viable for more than 15–20 years, and is able to germinate only in the presence of the host plant [5]. 

It is estimated that 16 million ha in the Mediterranean region and Western Asia are affected by broomrapes with worldwide annual crop losses as a result of broomrapes infestation about 1.17–2.33 billion € [6,7]. On average, sunflower seed losses caused by broomrape infestation can exceed 50% when susceptible hybrids are grown, frequently reaching 100% in heavily infested fields [8,9,10]. Considering the effects of parasite on crop production, different management and cultivation practices need to be used and/or combined. Chemical control methods for broomrape management are available, but high costs and environmental concerns limit their application. The use of broomrape-resistant cultivars is considered the most eco-friendly practical approach to control the parasite. Breeding for broomrape resistant cultivars includes exploitation of available and efficient sources of resistance, as an easy way to incorporate sources of resistance into existing material, and practical screening procedures to provide sufficient selection pressure. However, it requires continuous work, since as soon as resistance to the latest race is found, broomrape responds by evolving a more virulent race. Both qualitatively and quantitatively inherited resistances have been reported in sunflower against *O. cumana*. Qualitative resistance to broomrape is controlled by major genes and is race-specific [11]. Considering that resistance genes are rapidly overcome, it is necessary to search for new resistance sources and combine available genes [12].

This review focuses on genetics, genomics, and breeding of sunflower cultivars for improved resistance to broomrape, with special emphasis on conventional and molecular breeding efforts in creating sunflower cultivars resistant to more virulent broomrape races. It also deals with the structure and distribution of new physiological races, sources of resistance for introduction into susceptible sunflower cultivars, qualitative and quantitative resistance genes along with gene pyramiding and marker-assisted selection (MAS) strategies applied in the process of increasing sunflower resistance. In addition, it presents an overview of underutilized biotechnological tools, such as phenotyping, -omics, and genome editing techniques, which need to be introduced in the study of sunflower resistance to broomrape in order to achieve durable resistance.

## 2. Current Racial Status of Broomrape World-Wide 

Sunflower broomrape is host-specific, a trait rarely found among parasitic plants [13]. It is dominantly self-pollinated and, as a result, high genetic diversity is expected between populations. This was observed in regions with long history of broomrape occurrence [14]. Differences in intra-population genetic diversity were reported in the case of broomrape populations from Russia and Kazakhstan which were more diverse than the Romanian populations [15]. Contrary to that, broomrape introduction into new regions resulted in low genetic diversity and a probable connection to strong founder effect [16]. *O. cumana* is well adapted to agricultural systems and its disturbances resulting from cropping practices. Persistence of the broomrape seed bank is caused by the high production and longevity of seeds. 

*O. cumana* populations are commonly classified into races according to their virulence. Thus far, eight races have been reported, designated with letters from A to H [17]. Virulence discrimination of *O. cumana* was first reported as a difference between broomrape ability to attack varieties in the former Soviet Union, and races were described as A and the complex of races B [18]. Genetic resistance had ultimately become ineffective as the new race C emerged and severely affected sunflower production in Moldova [14]. Introduction of resistant sunflower germplasm provided parasite control for a certain period of time, but it was followed by the spread of more virulent *O. cumana* races. Currently, *O. cumana* is present in all the countries of Southern Europe and areas around the Black Sea where sunflowers are grown, North Africa, Israel, and China [10]. The presence of diverse broomrape races is reported in the southern regions of the Russian Federation, named F, G, and H [19]. In Kazakhstan, races C and G have been detected [18]. Broomrape races dominant in Ukraine during the 1980s and 1990s were race C and, to a lesser extent, race D. More virulent races E, F, and G were reported in sunflower cropping areas in Ukraine [20]. The presence of race H was observed in northeast of Ukraine, along with biotype with virulence higher than race H [21]. Results from Moldova indicated uneven spatial distribution of races, with less virulent races up to race E dominant in the central parts of the country, while more virulent races, with over 60% of broomrape samples, found in the southern and northern part of the country have been classified as races G and H [22]. In Romania, highly virulent broomrape race G appeared in 2005 [23], and races more virulent were confirmed in later research [24]. However, there is a significant difference in the distribution of virulence as the most virulent races are commonly found in regions near the Black Sea [25]. In 2003 and 2004, race F was detected in Bulgaria based on the susceptibility of the line P-1380 bearing gene *Or*_5_ and was found scarcely in the examined region [26]. In the period 2008–2017, permanent monitoring of broomrape virulence showed that the racial composition changed. The presence of races C, E, F, G, and H was determined, with the prevalence of races E and G and a decline in the presence of race F [27]. Race E is dominant in the area infected with broomrape in Serbia and race F was locally detected [28,29]. In the report of Molinero-Ruiz et al. [30] races commonly found in Spain were E and F. Recently, spreading of broomrape was recorded in the northern parts of Spain with a source in the Guadalquivir Valley, race F being the most virulent one [31]. At the same time, a change in broomrape virulence is also taking place in Spain, observed as broomrape occurrence on hybrids with resistance to race F, and labelled with sign G_GV_ [3]. This event was peculiar because a susceptible reaction of inbred lines DEB-2 and P-96 was lacking although resistance in commercial hybrids was overcome. Molinero-Ruiz et al. [30] were the first to report the presence of broomrape race F in Hungary and confirm existence of races F in Spain and Turkey, as well as race G in Turkey. Races E and G were determined in broomrape populations in Tunisia [32], and testing Moroccan *O. cumana* population indicated the presence of race G [33]. Mapping racial structure and distribution in China, presence of races A–G have been identified [34,35]. 

From the point of production sustainability, coexistence of *O. cumana* and its host in various environments often leads to virulence change. Determination and mapping of races revealed a high level of divergence along with a common trend of virulence increase. Broomrape attack on previously resistant sunflower genotypes results in labelling parasite populations with the same letter, indicating the need for comparative studies of populations in different countries [36]. Better understanding of processes which condition the emergence of new virulence biotypes and their comparison is therefore needed, together with fundamental research at the cellular level of the interaction between the host and parasite in the early developmental stages [37]. The results of conducted research on virulence increase are diverse in nature. Single point mutation was reported as a probable cause of race F emergence [38]. Gene flow from *O. cumana* from wild *Asteraceae* was also indicated as the potential source of virulence diversity, as well as the admixture of the local populations [3,39]. The mechanisms to overcome resistance may already exist in some broomrape individuals within a population [40]. The increase in virulence is considered as the consequence of host selective pressure. However, susceptibility of line NR5, resistant to race E, was recorded for broomrape populations collected in Spain before the introduction of hybrids resistant to race E [40]. Data obtained from research on sunflower broomrape reveals the complexity of interaction, with the ultimate goal of achieving long-term control of this parasitic weed.

## 3. Genetic-Based Improvement of Broomrape Resistance 

### 3.1. Sources of Broomrape Resistance 

The success of selection greatly depends on the presence of genetic variability for the traits concerned [41]. Considering the relatively narrow genetic base of cultivated sunflower, the existing genetic resources are an invaluable source of variability, which can be used for the introduction of agronomically important genes to improve the quality and economic value of sunflower crops. As for broomrape, wild *Helianthus* species are a most important source of resistance, but other sources, such as open pollinated varieties, and different gene pools of cultivated sunflower developed in research institutions around the world, have also been used for the introduction of broomrape resistance. 

**Open pollinated varieties**: The beginnings of sunflower breeding relied on local cultivars developed in the former Soviet Union. Local populations are of great importance as they possess many valuable genes, especially those addressing higher adaptability to specific environmental conditions and resistance to certain diseases. The first sunflower varieties resistant to broomrape were developed by the Soviet breeders in the first half of the 20th century. Those were local varieties, such as Saratovsky 169, Zelenka and Fuksinka, resistant to race A. Especially important are varieties resistant to broomrape, like Zhdanovsky 6432, Zhdanovsky 8281, and Stepnyak, created by academician Zhdanov and developed at the Saratov experimental station when the occurrence of broomrape race B threatened to jeopardize sunflower production [42]. One of the largest collections of open-pollinated varieties is maintained at the Vavilov All-Russian Institute of Plant Genetic Resources from Saint Petersburg, Russia containing 400 genotypes.

**Gene pool of cultivated sunflower**: The existing gene pools of cultivated sunflower are also an important source of broomrape resistance, especially considering that they are created under different conditions. These gene pools are composed of different types of synthetic populations and inbred lines created over many years of breeding. Resistance to broomrape in cultivated sunflower has started to be widely explored after the introduction of inbred lines and hybrids in the 1960s and 1970s. Vranceanu at al. [43] used the cultivated sunflower gene pool of the Agricultural Research and Development Institute from Fundulea, Romania to develop a set of sunflower differential lines for broomrape races from A to E, each line carrying a single dominant gene (*Or*_1_ through *Or*_5_, respectively) conferring resistance to the corresponding race. Pacureanu et al. [23,44] identified lines LC 1093, LC 009, and AO-548 resistant to races F and G from the same gene pool (Table 1). There are several reports of inbred lines resistant to race G of broomrape originating from the different cultivated sunflower germplasm collections: line HA 267 was selected from gene pool of Institute of Field and Vegetable Crops (IFVCNS), Novi Sad, Serbia, line LR1 was developed in the Institut National de la Recherche Agronomique (INRA) in Toulouse, France, while a group of lines resistant to race G was developed at the All-Russian Research Institute of Oil Crops by Pustovoit V.S. (VNIIMK), Krasnodar, Russia (Table 1). These, and other, institutes contributed considerably to enriching sunflower genetic resources by the development and improvement of sunflower genotypes resistant to various broomrape races.

**Wild sunflower species**: Unlike many other crops, a collection of wild sunflower species, specifically 53 wild species, of which 39 perennials and 14 annuals are at the disposal to sunflower breeders [45]. As sunflower wild species grow in a diverse range of habitats, i.e., plains, deserts, salt marshes, forests, and mountains, they are adapted to different environmental conditions and have considerable variability of biotic and abiotic resistance traits. One of the first uses of wild sunflower species is linked to the beginnings of sunflower breeding, when the Russian scientist Sazyperow tried to incorporate resistance to rust from *H. argophyllus*, while academician Zhdanov successfully used *H. tuberosus* for the development of cultivars resistant to broomrape [4,46]. Since then, wild sunflower species have been the primary genetic source of resistance to important diseases limiting sunflower production. The largest and the most important wild species collection of the genus *Helianthus*, as well as important collection of elite sunflower germplasm is the USDA-ARS National Plant Germplasm System (NPGS) maintained at the North Central Regional Plant Introduction Station (NCRPIS) in Ames, Iowa, USA [47]. Thanks to the efforts of researchers who exploited the collection from USDA-ARS there are now important bigger and smaller collections of sunflower wild species in other countries like in Serbia (IFVCNS), Bulgaria (Dobroudja Agricultural Institute, General Toshevo), France (INRA), Argentina (Instituto Nacional de Tecnología Agropecuaria, Pergamino), Spain (Institudo de Agricultura Sostenible, Cordoba), Ukraine (Institute of Oilseed Crops, Zaporozhie) and in Russia (Vavilov All-Russian Institute of Plant Genetic Resources, Saint Peterburg and Institute of Sunflower, Veidelevka). 

The resistance to broomrape in wild *Helianthus* species has been known since early sunflower breeding research in the former Soviet Union [48] to recent reports [45]. A breakdown of resistance through the incidence of new broomrape races forced breeders to search for different sources of resistance [49,50,51,52]. Different levels of broomrape resistance have been identified in some accessions of *H. tuberosus*, *H. grosseserratus*, *H. mollis*, *H. nuttalii*, *H. debilis*, *H. neglectus*, *H. niveus*, *H. argophyllus*, *H. petiolaris*, and *H. praecox* [53,54]. With the occurrence of more virulent broomrape race F, Jan et al. [55] employed interspecific hybridization to incorporate genes for resistance from several wild species into cultivated sunflower, thereby developing four resistant populations (BR1-BR4). Later, the population of wild sunflower species *H. debilis* was found to possess a dominant gene for resistance to the new race G [56]. A more recent study reports that dominant resistance to broomrape race classified as G had been detected in an interspecific cross between *H. annuus* and *H. debilis* subsp. *tardiflorus* [57]. Resistance to race G was also reported by Cvejić et al. [58] in a fertility restorer derived from *H. deserticola*. Resistance to races F and G in accessions of *H. praecox*, *H. debilis*, *H. petiolaris*, *H. tuberosus* and *H. maximiliani* was reported by Anton et al. [59]. Sunflower inbred line LIV-17 derived from the interspecific cross with *H. tuberosus* was found to carry recessive resistance to broomrape populations present in Turkey and Spain [60]. Additionally, new source of post-haustorial resistance to broomrape race G in *H. praecox* has been reported by Sayago et al. [61].

The transfer of genes for resistance to broomrape from wild sunflower species into cultivated sunflower is done by interspecies hybridization [2,28]. Introgression of resistance genes from wild species is not an easy task because many other unwanted (wild), agronomically undesirable traits, are introduced during this process. After the introduction of resistance genes, a large number of crossings and selections have to be carried out in order to improve important agronomic traits and maintain desired level of broomrape resistance at the same time (Figure 1). 

### 3.2. Resistance Genes and Resistant Sunflower Genotypes

Breeding for resistance is a continuous and extensive work, which includes discovering resistance gene(s) and development of resistant sunflower genotypes. Broomrape resistance genes are denoted as *Or* genes. Vranceanu et al. [43] identified five single dominant genes (*Or_1_–Or*_5_) for resistance to five races (A-E) of broomrape. They established the set of five differential lines resistant to five successive races. Inbred lines and hybrids resistant to race E were successfully developed, thus improving sunflower production in Europe, Asia, and worldwide. This type of resistance is simply referred to as qualitative or race-specific or gene-for-gene resistance [11]. It is highly efficient in complete parasite inhibition, and has become preferable among plant breeders due to the simplicity of selection in breeding programs [62]. However, this type of resistance can easily be broken down due to rapid evolution of the pathogen [41,63].

Resistance was overcome by the appearance of new races, however, it mostly remains race-specific. The appearance of race F led to identification of resistant genotypes controlled by a single dominant gene *Or*_6_ [64,65], two recessive genes *or*_6_*or*_7_ [66,67], and two partially dominant genes *Or*_6_*Or*_7_ [68] (Table 1). Thus, the difference in the mode of inheritance resulted from different backgrounds of the genetic material. The dominant *Orobanche* resistance genes have previously been used in sunflower breeding programs, until the emergence of recessive resistance for the first time. Recessive genes could impact more achieving durable resistance against the respective parasite, but brought about the need to incorporate resistant genes into both parental lines for developing resistant hybrid [66]. Most models that attempt to explain the resistance suggest that dominant resistance is an active process, where the plant synthesizes compounds that interfere with the parasite. Conversely, recessive resistance might be the result of plant cells lacking some factor(s) essential for the pathogen’s life cycle [12]. When new, more virulent populations (G and H) arose, several new sources of resistance have been reported, indicating both dominant and recessive inheritance (Table 1). 

In addition to the studies on qualitative resistance, recent genetic and molecular studies have revealed a more complex control of broomrape resistance in sunflower. The race-specific resistance to *O. cumana* have been reported for quantitative loci [65,68,72,74] (Table 1). Using different mapping populations, authors have mapped a QTL-controlling resistance to broomrape in different genetic regions. The main advantage of the approach is that, besides major QTL, there are complementary QTL with minor effects on broomrape resistance, which can be used as donor sources for marker-assisted pyramiding programs. According to Pérez-Vich et al. [65], the contribution of QTL to broomrape resistance has not always corresponded to alleles from the resistant parent. In some cases, it was conferred by alleles of the susceptible parent.

The main objective of sunflower breeding for broomrape resistance is to develop high-yielding sunflower genotypes, carrying desired *Or* genes. Genotypes identified for resistance to highly virulent races of broomrape represent a valuable source for gene transferring and can be used for racial differentiation. There is no universally accepted set of differential lines for identification of races over F and they are specific to individual seed companies or research groups [3,82]. Most often, breeders use inbred lines from previous studies for racial classification (LC-1093, P96, HA267, DEB-2, etc.) but broomrape populations in some particular areas are insufficiently distinguished. The lack of corresponding differential inbred lines caused the extensive use of broomrape-resistant commercial hybrids (Transol, ES Bella, LG-5580, PR66LE25, and others) as standards throughout Europe. Generally, hybrids exhibit better and more durable resistance to broomrape compared to inbred lines, due to the combination of resistance mechanisms. For research purposes, it is essential to have a set of differential lines accessible to everyone in order to determine the virulence of particular broomrape populations.

## 4. Genomic-Based Improvement of Broomrape Resistance 

Broomrape resistant genotypes with incorporated single *R* gene often lose their resistance in a very short period. It occurred when sunflower commercial hybrids carrying *Or*_5_ gene lost their resistance to races more virulent than E. Sunflower breeding for sustainable broomrape resistance therefore needs new strategies, such as pyramiding of major genes or combining qualitative and quantitative resistance mechanisms [11]. Pyramiding of more than one gene and QTL into a single genetic background deteriorate parasite to overcome two or more resistance genes, compared to the one controlled by only one single gene. However, gene pyramiding through traditional breeding is difficult to achieve due to linkage drag, which is often difficult to break even after several back crossings [41]. Gene pyramiding through MAS is, therefore, a more effective approach for bringing rapid genetic improvement (Figure 2).

### 4.1. Molecular Markers for Identification of Major and Minor Genes Involved in Broomrape Resistance

Marker-assisted selection (MAS) offers a simple, more efficient, accurate breeding method, convenient in breeding for disease resistance compared with selection based on phenotype [83]. As broomrape resistance is introduced into cultivated sunflower from various sources, the origin of the material is an important factor when it comes to mode of inheritance and the position of resistance genes. Different types of molecular markers have been employed for the localization and identification of region(s) carrying resistance genes. With the emergence of more complex, quantitative broomrape resistance in sunflower, analyzing and identifying genes for resistance has become more challenging, requires the use of more sophisticated tools. 

Several broomrape resistance genes have been mapped so far. Gene *Or*_5_ has been mapped on the telomeric region of chromosome 3 [84,85,86]. Furthermore, Pérez-Vich et al. [65] detected five QTL for resistance to race E and six QTL for resistance to race F, in seven different chromosomes. Phenotypic variance for race E resistance was mainly explained by the major QTL *or3.1*, on chromosome 3, associated to the resistance or susceptibility character. Markers for genes conferring resistance to races higher than F have been reposted [12,73]. The closest marker, tentatively designated as or_ab-vl-8_, was ORS683 with the genetic distance of 1.5 cM on chromosome 3 [10]. Although this gene was mapped on the same chromosome as *Or*_5_, the authors proved that the two resistance genes are different. While *Or*_5_ gene resides in the upper terminus of chromosome 3 with the closest public marker ORS1036 being 7.5 cM downstream, *or_ab-vl-8_* is mapped in the lower region of chromosome 3. Later on, Imerovski et al. [74] mapped two major QTL on chromosome 3, designated QTL *or3.1* and QTL *or3.2*. QTL *or3.1* was positioned in a genomic region where the previous broomrape resistance gene *Or*_5_ had been mapped, while QTL *or3.2* was identified for the first time in the lower region of the same chromosome. The authors analyzed four different crosses with four different broomrape resistance sources, and they were, therefore, able to identify major and minor QTL and found that chromosome 3 carried resistance QTL in all crosses. Depending on the cross, between two and 23 significant QTL were mapped across the sunflower genome. Furthermore, CAPS markers have been developed for facilitation of introgression of major QTL in the region of the peak for *or3.2* which can be useful in sunflower breeding for resistance to race G. Another example of recessive resistance was reported by Akhtouch et al. [68] who detected QTL for recessive resistance to broomrape race F. However, fine mapping is needed in order to identify markers that could be used in MAS.

Molecular markers have been patented for the QTL conferring broomrape System 2 resistance [79]. This type of resistance enables broomrape to germinate and form the haustorium, but the parasite dies before emerging from the soil and completing its life cycle. A putative locus for broomrape race H System 2 resistance was mapped to LG4 of the SSR map, approximately 3 cM from the SSR marker HT0664-CA. Two SNP markers (HT0183 and HT090) were identified as potential markers of interest. In the study concerning the characterization of the HA89xLR1 population, Louarn et al. [72] identified different QTL for each race (F from Spain and G from Turkey) and each of the three stages of broomrape development. In total, 17 QTL were mapped on nine different sunflower chromosomes, with only one QTL co-localizing for resistance for both races. Recently, *HaOr*_7_ gene, a major resistance gene that confers resistance to *O. cumana* race F, has been mapped on chromosome 7 [81]. A major resistant QTL marker for *Or_Deb-2_* gene have been mapped, which explained 64.4% of the total phenotypic variation, on chromosome 4 and other QTL with minor effects [77].

### 4.2. Association of Identified QTL with Resistance Genes and Functional Genomics

Histological and physiological studies of host resistance mechanisms to broomrape infection established the basis for molecular studies of resistance. Different modes of resistance were reported, such as no or low stimulation of germination of broomrape seeds, necrosis of the parasite, and different histological changes in the host to disrupt the formation of a functional connection between the parasite and the host plant [87]. Early work on detecting genetic mechanisms of sunflower resistance revealed that sunflower reacts to the presence of broomrape seedlings within the first 2 h. Overall, transcript accumulation of 11 defense-related genes was highlighted by Letousey et al. [87]. Two major resistance mechanisms were revealed in LR1 genotype: callose accumulation correlated with over expression of *HaGSL1* and induction of *def*. gene. Further on, Louarn et al. [72] who also used LR1 line as the resistant parental line in their study, identified one QTL on chromosome 15 collocating with a *NBS-LRR* (Nucleotide-binding site leucine-rich repeat) gene, previously identified by Radwan et al. [88] using candidate gene approach. Moreover, BlastX analysis of the sequence of genes involved in cowpea-Striga resistance [89] showed the presence of three homologous genes in the sunflower genome. Genes homologous with the cowpea gene, predicted to belong to the coiled-coil-nucleotide-binding-site-leucine-rich repeat (*CC-NBS-LRR*) gene family, were found to collocate with QTL on chromosomes 13, 15, and 17. These QTL were found to control broomrape field emergence (*HaT13l034464*, chromosome 13), the capacity to induce incompatible attachments (*HaT13l008311*, chromosome 15), and induce necrotic tubercle (*HaT13l008327*, chromosome 17). Recently, Şestacova et al. [76] tested seven resistant, tolerant, and susceptible sunflower lines and reported higher stability in transcriptional activity of four examined defense genes: *NPR1, PAL, defensin* and *PR5* in resistant genotypes. The results showed that resistant genotypes were able to maintain and recover their normal level of metabolism when exposed to stress conditions.

Recent biotechnological advancements in analysis of the whole genome, proteome, and metabolome of plants, enabled the discovery and characterization of multiple genes/proteins involved in resistance on a wider scale. In sunflower, the solid basis for exploitation of sunflower genome was made with the recent publicly available genome sequence [90]. Duriez et al. [81] combined genomics and map-based cloning strategy in order to map and identify the function of *HaOr*_7_ gene conferring resistance to race F. The gene was located in a window of around 55 kb. To obtain the parental genomic sequences, the authors created and screened two BAC libraries from the susceptible and resistant lines. The comparison of both genomic sequences showed a high level of divergence with large structural variations, suggesting a wild origin of the *HaOr*_7_ gene. After comparing the sequence with the reference sequence of XRQ line susceptible to race F, the authors were able to determine that the *HaOr*_7_ gene encodes a complete receptor-like LRR kinase protein while in the susceptible sunflower line, this protein is truncated lacking the transmembrane and the kinase domains. *HaOr*_7_ prevents the formation of broomrape connection to the vascular system of sunflower roots in a gene-for-gene relationship. This gene acts at the early response stage.

Analyzing three different crosses, Imerovski et al. [74] exploited the regions on chromosome 3 in the intervals where QTL peaks were found to be significant. The author identified 123 genes in the region of major QTL *or3.1*, between 31.9 and 38.48 Mb. Among those two were singled out: *HanXRQChr03g0065701* (disease resistance protein RPS2-like) and *HanXRQChr03g0065841* (TMV resistance protein Nlike) (www.heliagene.org). Furthermore, 71 genes were identified in the region of QTL *or3.2*, between 97.13 and 100.85 Mb, including a putative defense gene *HanXRQChr03g0076321*. Considering that the region surrounding QTL *or3.1* had previously been identified as carrying gene *Or*_5_ conferring complete resistance to race E and partial resistance to race F [65,86], the authors hypothesized that region QTL *or3.1* could in fact be *Or*_5_ gene. Since Imerovski et al. [74] exploited resistance genes/QTL conferring resistance to race G, it may happen that *Or*_5_ now acts as the so called “defeated *R* gene”, which expresses moderate resistance to races higher than race E. This effect is observed in other crops as well [91,92] in which genes conferring complete resistance toward a race/strain of pathogen, express partial resistance toward a more aggressive strain/race of the same pathogen. 

The first study by Yang et al. [93] provided an overall insight into compatible and incompatible interaction mechanisms between resistant and susceptible sunflower genotypes and broomrape. Over 3500 proteins were identified in tested sunflower genotypes by iTRAQ analysis. Response of the resistant genotype to broomrape race G infection included regulation of pathways associated with energy metabolism, alteration of defense-related proteins that participate in the recognition of the parasite, accumulation of pathogenesis-related proteins, lignin biosynthesis, and detoxification of toxic metabolites. In the susceptible genotypes, the decrease was observed considering the abundance of proteins involved in the biosynthesis as well as the signaling of plant growth regulation [93]. In this study, the molecular basis of sunflower resistance to broomrape has been broken down into steps and explained in detail.

## 5. Future Prospects and New Approaches

The review provides a comprehensive overview of the efforts made by the international scientific community in controlling broomrape, with an emphasis on the recent events. As previously stated, there is a constant and dynamic relationship between sunflower and broomrape. Sunflower breeders constantly attempt to exceed this apprehensive parasitic weed. Successful control of broomrape requires a trans-discipline integrated approach, starting from better exploitation of genetic resources by modern phenotypic tools, through exploring -omics techniques, such as genomics, transcriptomics, metabolomics, and epigenomics, and combining the obtained data (Figure 2).

Technological development will certainly help improve the conventional methods of broomrape control, allowing more precise assessment of larger panels of genotypes. One of the solutions to accelerate broomrape resistance could be image-assisted phenotyping. Similar to human screening for body temperature to detect infections at airports, thermal imaging allows the detection of infected sunflower plants due to an increase in leaf temperature as a consequence of parasite-induced stomatal closure and reduction of transcription [94]. Moreover, hyperspectral imaging allows the detection of early-stage broomrape infestation, as the parasite causes quick changes in nutrient content of leaves which immediately leads to changes in the leaf mesophyll [95]. These and other similar methods in combination with powerful statistical tools for image analysis are not only helpful in terms of fast screening of large gene pools, but also in increasing efficiency of chemical control of broomrape. 

In sunflower, there are several major gene banks accessible for screening of resistance. There are over 15,000 different accessions of cultivated and wild sunflowers available for examining worldwide. It may even be hypothesized that resistance to the present and emerging races already exists in sunflower, because novel resistance sources are usually found within the existing sunflower genetic variability, as stated by Molinero-Ruiz et al. [40]. The presence of multiple resistance genes may offer greater evolutionary impedance than a single resistance gene, since a pathogen would have to develop mutations in all of the effectors that are recognized by the resistance genome complement in order to overcome complex resistance [96]. Furthermore, defense mechanisms of sunflower against the new, more virulent broomrape races became more complex. This requires application of new and high throughput biotechnological tools. Another setback of improving dominant resistance in sunflower is that it does not provide the long-lasting solution for broomrape control. Breeders pay more attention to gene pyramiding of not only major, but also the minor genes. The solution could be found in pyramiding “defeated *R* genes” together with new genes conferring resistance to more virulent races. Thus, breeders should carefully combine resistance genes that differ in mode of inheritance with the stage in which they confer resistance. 

Recent sunflower genome assembly will definitively help identify the possible candidate genes involved in resistance to broomrape as well as their function. Thus far, only a couple of authors have exploited the sunflower genome sequence in their molecular research, as is the case with the exploitation of various -omics techniques in examining sunflower-broomrape interaction. The progress in -omics techniques and powerful statistical tools in big data analysis should be exploited to the fullest for conducting more advanced and detailed research in revealing mechanisms underlying complex interaction between sunflower and broomrape as well as characterizing resistance pathways in sunflower. Unfortunately, so far there have been no reports on the examination of epigenetic mechanisms in sunflower resistance. As a new field of research, it would be very useful to examine to what extent epigenetic mechanisms influence resistance in sunflower, considering that the DNA methylation status plays a crucial role in regulating *Phelipanche ramosa* seed germination during conditioning period, by controlling the strigolactone-dependent expression of *PrCYP707A1* [97].

Moreover, sunflower broomrape genome was recently sequenced with the addition of 20 transcriptomic experiments for the annotation of the genome sequence [98]. This can be a starting point for identification of avirulence genes in broomrape as interactors with virulence genes in sunflower, a better insight into resistance mechanisms developed by sunflower as well as unravelling new resistance genes.

Developments in gene editing techniques could lead to the quicker design of superior broomrape resistant genotypes. Powerful CRISPR-Cas9 technique was successfully used for mutagenesis of the *CCD8* (Carotenoid Cleavage Dioxygenase 8) gene, strigolactone-biosynthesis gene, in order to create *Phelipanche aegyptica* resistant tomato lines [99]. Another gene silencing technique, virus-induced gene silencing (VIGS) was used to induce trans-silencing of *PaCCD7* and *PaCCD8* genes in *P. aegyptica* for significant reduction in the number of parasite tubercles attached to *Nicotiana benthamiana* roots [100]. New gene editing techniques can be difficult to apply in sunflower breeding, mainly due to the difficulties that occur during plant regeneration and low numbers of obtained transgenic regenerants per assay. Thus, the first step for use of the modern gene editing techniques would require the establishment of improved basis for transformation, which could be beneficial in the development of durable broomrape resistance in sunflower.

As previously stated, there are numerous approaches and techniques that can be exploited in achieving sustainable broomrape resistance in sunflower. However, the solution does not lie in one approach, but in combining multiple approaches and developing statistical tools for proper exploitation. Successful control of sunflower broomrape will, therefore, result from mutual efforts made by pre-breeders, breeders, phytopathologists, biotechnologists, and bioinformaticians.

## Figures and Tables

**Figure 1 genes-11-00152-f001:**
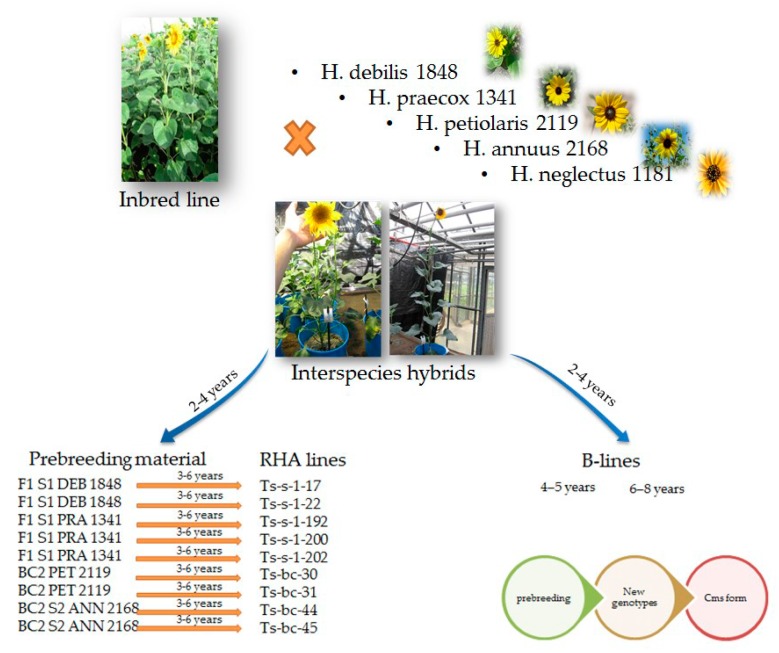
Example of introgression of resistance genes in cultivated sunflower genotypes from different accessions of wild *Helianthus* species.

**Figure 2 genes-11-00152-f002:**
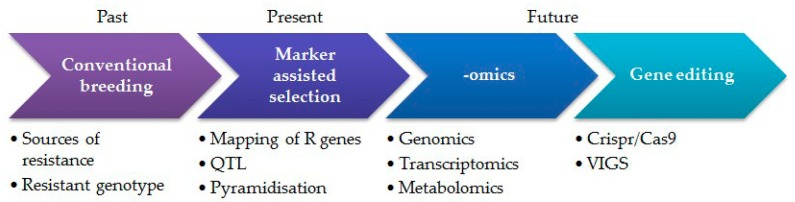
Integrated approach of sunflower breeding to broomrape resistance.

**Table 1 genes-11-00152-t001:** Genetic material and their inheritance of broomrape resistance to races F and G in sunflower.

Genotype Name	Source	Resistant to Race(s)	Gene(s)	Inheritance	Reference
R-96	Cultivated sunflower (Yugoslav origin)	F (Spain)	*or*_6_,*or*_7_	Two recessive genes	Fernandez-Martinez et al. (2004) [67]; Akhtouch et al. (2002) [66]
L-86	Cultivated sunflower (Russian origin)	F (Spain)	*or*_6_,*or*_7_	Two recessive genes	Fernandez-Martinez et al. (2004) [67]; Akhtouch et al. (2002) [66]
K-96	Cultivated sunflower (Russian origin)	F (Spain)	QTL	Recessive	Fernandez-Martinez et al. (2004) [67], Akhtouch et. al (2016) [68]
KI-534	unknown	F (Spain)	*or*_6_,*or*_7_	Two recessive genes	Rodriguez-Ojeda et al. (2001) [69]
BR-4 (J1)	Interspecies hybridisation (*H. grosseserratus* and *H. divarticatus*)	F (Spain)	*Or*_6_; *Or*_6_,*Or*_7_	Single dominant gene; Two partially dominant genes	Jan et al. (2002) [55]; Rodriguez-Ojeda et al. (2001) [69]; Velasco et al. (2007) [70]
P-96	Cultivated sunflower (Yugoslav origin)	F (Spain)	*or*_6_,*or*_7_	Two recessive genes QTL	Fernandez-Martinez et al. (2004) [67]; Akhtouch et al. (2002) [66]; Perez-Vich et al. (2004) [65,71]; Akhtouch et. al (2016) [68]
LC-1093	Cultivated sunflower	F (Romania)	*Or* _6_	Single dominant gene	Pacureanu Joita et al. (1998) [64]
AO-548	Inbred line from germplasm collection of Fundulea Institute	G (Romania)	unknown	Two independent dominant genes	Pacureanu Joita et al. (2008) [44]
LC-009	Inbred line from germplasm collection of Fundulea Institute	G (maybe new)	unknown	unknown	Pacureanu Joita et al. (2009) [23]
LR1	INRA	F (Spain) G (Turkey)	QTL	-	Louarn et al. (2016) [72]
HA267	selected from the Novi Sad gene-pool	G (Spain, Romania, Turkey)	unknown	Single recessive gene, QTL	Imerovski et al. (2014) [73]; Imerovski et al. (2019) [74]
AB-VL-8	interspecific hybridization with *Helianthus divarticatus*	G (Spain, Romania, Turkey)	*or_ab-vl-8_*	Single recessive gene, QTL	Cvejic et al. (2014) [75]; Imerovski et al. (2016; 2019) [16,74]
LIV-10; LIV-17	interspecific hybridization with *Helianthus tuberosus*	G (Spain, Turkey)	unknown	Single recessive gene, QTL	Cvejic et al. (2014; 2018) [60,75]; Imerovski et al. (2019) [74]
MS-2161A	created by AMG-Agroselect Company	G (Romania, Moldova)	unknown	unknown	Şestacova et al. (2016) [76]
DEB- 2	*H. debilis* subsp. *tardiflorus* (PI468691)	G (Spain)	*Or_Deb-2_*	Single dominant gene, QTL	Velasco et al. (2012) [57]; Gao et al. (2019) [77]
*H. praecox*	Provided from USDA-ARS	G (Posthaustorial resistance)	*Or_pra1_*	Single dominant gene	Sayago et al. (2018) [61]
VIR-665 VIR-221 VIR-222 No. 667 No. 769 No. 3046	VIR collection	G (Russia)	unknown	Single gene, incomplete dominance	Guchetl et al. (2018) [78]
PHSC1102	Pioneer Hi-Bred	F_GV_, G_TK_, G_RO_, and G_RU_	*Or_SII_*	Partial dominance	Hassan et al. (2008) [79]; Martin-Sanz et al. (2019) [80]
LSS, S and LSR	Syngenta Seeds	F (Spain)	*HaOr* _7_	Dominance	Duriez et al. (2019) [81]
PHSC0933	Pioneer Hi-Bred	F_GV_, G_TK_, G_RO_, and G_RU_	*Or* _7_	Dominance	Martin-Sanz et al. (2019) [80]

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
