# Peer review of "Genetic and Genomic Tools in Sunflower Breeding for Broomrape Resistance"

_genes, 2020, doi:10.3390/genes11020152_

Round 1

Reviewer 1 Report

It is a very interesting review. It was a pleasure to read and learn useful things for my own research. This is a very up to date work presenting the full story of the fight between sunflower and the broomrape. It is especially important now since the new technologies will definitely contribute to this research soon.

I still have some minor comments.

1) I would include some economic aspects to the introduction. It's important to show how beneficial it can be to simplify the resistance introduction into the sunflower lines since broomrape makes a big impact on sunflower production.

2) I think that some efforts have to be done to make the text more coherent, clear and consistent.

Section 2:  I would introduce the definition of the race for the broomrape before starting to describe them

Section 3.1 You have to improve the structure. Name the collections then point out the importance of them or the other way around. Now ir a bit messy

Section 3.2 and 3.3

3.2 has Or genes n the title but you explain what are these genes mean just in 3.3.

I think it is better to restructure the section 3. Because now the difference between the subsections is not clear. Very difficult to read. I would first describe the importance of wild lines in terms of resistances, then describe the existent collections, then will expand all about resistances to different races.  

Author Response

Dear Sir/Madam,

Thank you very much for your comments and useful suggestions. We appreciate your time spending for revision.

According to your comments, we are returning the manuscript with the rewriting of some parts. We attached manuscript using "Track changes" but when you accept all changes it is easier to reading.  

Here are explanations of main changes as response to your comments:

1) We added economic losses due to broomrape effect and emphasize the importance of sunflower breeding for broomrape resistance

2) In section 2 we added definition of races.

We rearranged Section 3:

We merged 3.1 and 3.2 and renamed Sources of broomrape resistance.  In this part we will present all available sources and exiting collections, with emphasis on wild sunflower species resistance to broomrape.

You can see detailed in the Manuscript.

Also, we renumbered references, so it looks messy now but when you read  version with accepted changes, it looks all right. 

We hope that the revised version of manuscript satisfy your demands and look forward to your response.

Best regards,

Cvejić et al.

Reviewer 2 Report

The paper “Genetic and genomic tools in sunflower breeding for broomrape resistance” by Sandra Cvejic et al. is a review on genetics, genomics and breeding of sunflower cultivars for resistance to broomrape. The manuscript is appears to make an interesting review on important aspects of breeding for broomrape resistance which constitutes one of the important objectives of sunflower breeding programs around the world.

In relation to literature cited and list of references there are some deficiencies that must be taken into account. The references are ordered by numbers and in some cases when they make a citation include only this number. For example, in the line 216 they write “historically (58) identified…” . I think that in these cases would be better to include the authors and the numbers of the list of references, for example “ Cvejic et al. (58). This occur several times in the text as in line 247 “ According to (67) … that should be changed by According to Perez-Vich et al. (67).. There are several more cases in lines 286, 294 and others.

Another deficiency is found in Table 1 where they include the authors but not the numbers. I think they should include both in order to find easily the reference in the list. For example, Akhtouch et al. 2002 (64), Perez-Vich et al. 2004(63). I do not find in the list the first reference in the table Fernández-Martínez et al. (2000). I think that these lines were registered in Crop Science in 2004.  The reference number 71 is not correct.

In conclusion as this paper is a revision the authors must check carefully the citations and the reference list.

Author Response

Dear Sir/Madam,

Thank you very much for your comments and useful suggestions. We appreciate your time spending for the revision.

According to your comments and comments of another reviewer, we are returning the manuscript with the rewriting some parts of manuscript. Some sections were rearranged. We used "Track change" but for easier reading we recommend to accept all changes.

In some cases we added name of first author due to easier reading as you requested. 

Also, we added numbers to all references in Table 1. Now all references are listed in the Reference section. 

All references were checked carefully and I hope it will not be incorrect ones. 

So, we renumbered references, it looks messy now but when you accept all the changes it will be all right.

We hope that the corrected version of manuscript satisfy your demands and looking forward to your response.

Best regards,

Cvejić et al.
